# Long-range transport of 2D excitons with acoustic waves

Ruoming Peng [1], Adina Ripin[2], Yusen Ye[3], Jiayi Zhu[2], Changming Wu[1], Seokhyeong Lee[1], Huan Li [1,5], Takashi Taniguchi [4], Kenji Watanabe [4], Ting Cao [3], Xiaodong Xu[2,3] & Mo Li [1,2 ✉]

Excitons are elementary optical excitation in semiconductors. The ability to manipulate and transport these quasiparticles would enable excitonic circuits and devices for quantum photonic technologies. Recently, interlayer excitons in 2D semiconductors have emerged as a promising candidate for engineering excitonic devices due to their long lifetime, large exciton binding energy, and gate tunability. However, the charge-neutral nature of the excitons leads to weak response to the in-plane electric field and thus inhibits transport beyond the diffusion length. Here, we demonstrate the directional transport of interlayer excitons in bilayer $WSe_2$ driven by the propagating potential traps induced by surface acoustic waves (SAW). We show that at 100 K, the SAW-driven excitonic transport is activated above a threshold acoustic power and reaches 20 μm, a distance at least ten times longer than the diffusion length and only limited by the device size. Temperature-dependent measurement reveals the transition from the diffusion-limited regime at low temperature to the acoustic field-driven regime at elevated temperature. Our work shows that acoustic waves are an effective, contact-free means to control exciton dynamics and transport, promising for realizing 2D materials-based excitonic devices such as exciton transistors, switches, and transducers up to room temperature.

[1] Department of Electrical and Computer Engineering, University of Washington, Seattle, WA 98195, USA. [2] Department of Physics, University of Washington, Seattle, WA 98195, USA. [3] Department of Material Science and Engineering, University of Washington, Seattle, WA 98195, USA. [4] Research Center for Functional Materials, National Institute for Materials Science, Tsukuba, Japan. [5] Present address: Zhejiang University, Hangzhou, China. ✉email: moli96@uw.edu

Excitons in semiconductor systems can be optically excited and read out, thereby encoding and storing optical signals into the excitons' spin, valley, and orbital degrees of freedom[1–4]. In analogous to electronic circuits, circuits with excitons as the active information carriers have been envisioned, which transport and manipulate excitonic states with applied electrical and magnetic fields[5,6]. Transducing between photons and solid-state media, such excitonic circuits can directly process optical signals and regenerate light without additional optical-electrical conversions so that they can be very efficient[6]. However, unlike electrons or holes, charge-neutral excitons experience no net force under uniform electric fields. They can also be dissociated by a moderately strong in-plane electric field if the binding energy is small, for example, in GaAs quantum well systems[7,8]. Therefore, diffusion has been one of the main mechanisms utilized for exciton motion in GaAs[9,10] and 2D materials[11–13]. Particularly, in GaAs systems, the high mobility of exciton has enabled 10 s of microns diffusion length at low temperature[6]. To actively transport excitons with controlled directionality, surface acoustic waves (SAW) have been employed to effectively transport excitons in GaAs quantum wells[14–18]. Over 100 s of microns transport distance is achieved at a temperature below < 4K[15,16]. However, the low exciton binding energy of only a few meVs in GaAs prohibits operation at higher temperatures[19].

Compared to GaAs quantum wells, excitons in 2D transition metal dichalcogenides (TMDCs) (e.g., $MoS_2$, $MoSe_2$, $WS_2$, $WSe_2$) have binding energy on the order of hundreds of meVs[3,4] with strong resonances even at room temperature (RT)[20–23], making them promising for a plethora of optoelectronic and quantum applications[24–28]. Particularly, the indirect excitons (IXs) in bilayers and heterobilayers of TMDCs have additional desirable properties[27,29–32]. Because these IXs consist of electrons and holes separated in different layers and valleys, their population lifetimes at low temperature are up to 100 s of nanoseconds[4], facilitating long-range transport before relaxation. Importantly, IXs have a permanent perpendicular dipole moment so that their energy can be tuned with an out-of-plane electric field[33,34], thus they can be driven by a lateral gradient of electric-field[35]. Indeed, prototypical TMDC excitonic transistors have been demonstrated based on field-controlled IX diffusion in $MoSe_2/WSe_2$ heterobilayers[35–37]. The application of an out-of-plane static electric field creates an energy barrier or trap for IX, so switching on and off the electric field can suppress or enable exciton diffusion[37]. However, due to limited exciton mobility[38], diffusive and repulsive transport of IXs can only achieve a transport distance of a few µm using milliwatts of optical excitation power at a low temperature[39]. More importantly, such diffusive transport is non-directional, thus challenging to realize functional excitonic circuits, which entail transporting excitons in a controlled direction over a long distance.

## Results

Here, we demonstrate the long-range and directional transport of IXs in a bilayer $WSe_2$ using SAW at temperatures up to RT. The IXs in a bilayer $WSe_2$ are momentum indirect with a long lifetime of up to 10 ns at 10 K and ~1 ns even at RT[29,37]. It was reported recently that the exciton energy of IXs in bilayer $WSe_2$ could be modulated by an out-of-plane static field[29]. Consider a bilayer $WSe_2$ placed directly on a piezoelectric substrate (Fig. 1a). SAW is excited and propagates through the area where a bilayer $WSe_2$ is transferred to. On a piezoelectric substrate such as $LiNbO_3$, the propagating SAW will generate a near-field piezoelectric field with a large out-of-plane field ($E_z$) amplitude on the order of $10^7$ V/m at 1 mW/µm of acoustic power density. As such, the $E_z$

component will periodically modulate the IX energy in space and time, creating a dynamic trapping potential for the IXs in the extrema of $E_z$. As the SAW propagates, the IXs will drift along the time-varying gradient of $E_z$ and thus be carried by the SAW—like surfing on a wave—to travel a long distance before they recombine (Fig. 1b). With a SAW velocity of ~$3.0 \times 10^3$ m/s and an exciton population lifetime > 10 ns, the IXs can travel >30 µm, an order of magnitude longer than the exciton diffusion length.

Figure 1b depicts the scenario when a bilayer $WSe_2$ is under both optical pumping and SAW modulation. Since the bilayer $WSe_2$ is inversion symmetric, there are two energy degenerate IXs with dipole moment $p$ pointing along the $+z$ and $-z$ directions, respectively. For simplicity, hereon, we use $+z$ IX to explain the transport process. The electric field $E(r,t)$ induced by the SAW modulates the IX energy by $\Delta U = -p \cdot E$ and thus creates a dynamic potential well moving at the acoustic velocity. At low temperatures, the optically excited IXs will quickly relax to the energy minimum created by the SAW in real space and travel with the propagating SAW. Figure 1c shows the optical image of our device. We designed a focusing IDT structure that focuses the acoustic wave into the $WSe_2$ region (Fig. 1d) to concentrate the acoustic power density and enhance the piezoelectric field $E$. The IDT excites a strong SAW mode at 1.237 GHz with an acoustic wavelength of 2.832 µm, assuming an acoustic velocity of $3.5 \times 10^3$ m/s for the z-propagating Rayleigh mode in a y-cut $LiNbO_3$ substrate (see S.I.). Figure 1d shows the simulated electric field profile of the focused beam of the acoustic wave, which at the focal point has a waist of ~3.0 µm. To reduce the inhomogeneous broadening and exciton trapping from spatial variation of surface potential, we encapsulated the bilayer $WSe_2$ with ~10 nm hexagonal boron nitride (h-BN) flakes using the standard pick-up method and transferred onto the $LiNbO_3$ substrate with pre-patterned IDTs[40]. A thin layer of indium-tin-oxide (ITO) was deposited on the heterostructure region as a top transparent electrode. This top electrode plays an important role by efficiently suppressing the in-plane piezoelectric field component ($E_x$), which may cause exciton dissociation[14,41–43], while maintaining a relatively strong out-of-plane field component $E_z$. The finite-element method (FEM) simulation (Fig. 1e, f) compares the piezoelectric fields in the situations of with (Fig. 1e) and without (Fig. 1f) the top ITO layer, showing that the ITO layer suppresses $E_x$ by about two orders of magnitude.

We first demonstrate efficient SAW transport of IXs by performing spatially resolved photoluminescence (PL) measurement. Without SAW and at a low pump power, the diffusive IX flux density can be described by the diffusion equation: $j = -D\nabla N$, where $D$ is the temperature-dependent diffusion coefficient and $N$ is the exciton density (inset, Fig. 2a). Due to the low $D$ in TMDC, the IX diffusion is relatively weak. In comparison, when SAW is turned on, the IX population is strongly modulated by the piezoelectric field and drifts along the SAW propagation direction (inset, Fig. 2b). As the velocity of the SAW (~3,500 m/s) is much lower than the thermal velocity of excitons ($10^4 – 10^5$ m/s in our experimental condition, see S.I.), the IXs can be treated as an exciton gas in quasi-equilibrium. With sufficiently high SAW amplitude, the IX gas will be trapped in the energy minimum of SAW and drift with a center-of-mass velocity identical to the SAW velocity.

**Interlayer exciton transport**. Figure 2a, b compare the PL images measured at 100 K with SAW off and on, respectively. The IXs were excited with a He-Ne laser at 633 nm, having a power of $P_p = 20$ µW, and focused to a diffraction-limited spot size of 1 µm. The excitation spot is placed near the edge of the $WSe_2$ flake close to the IDT and in the middle of the acoustic wave

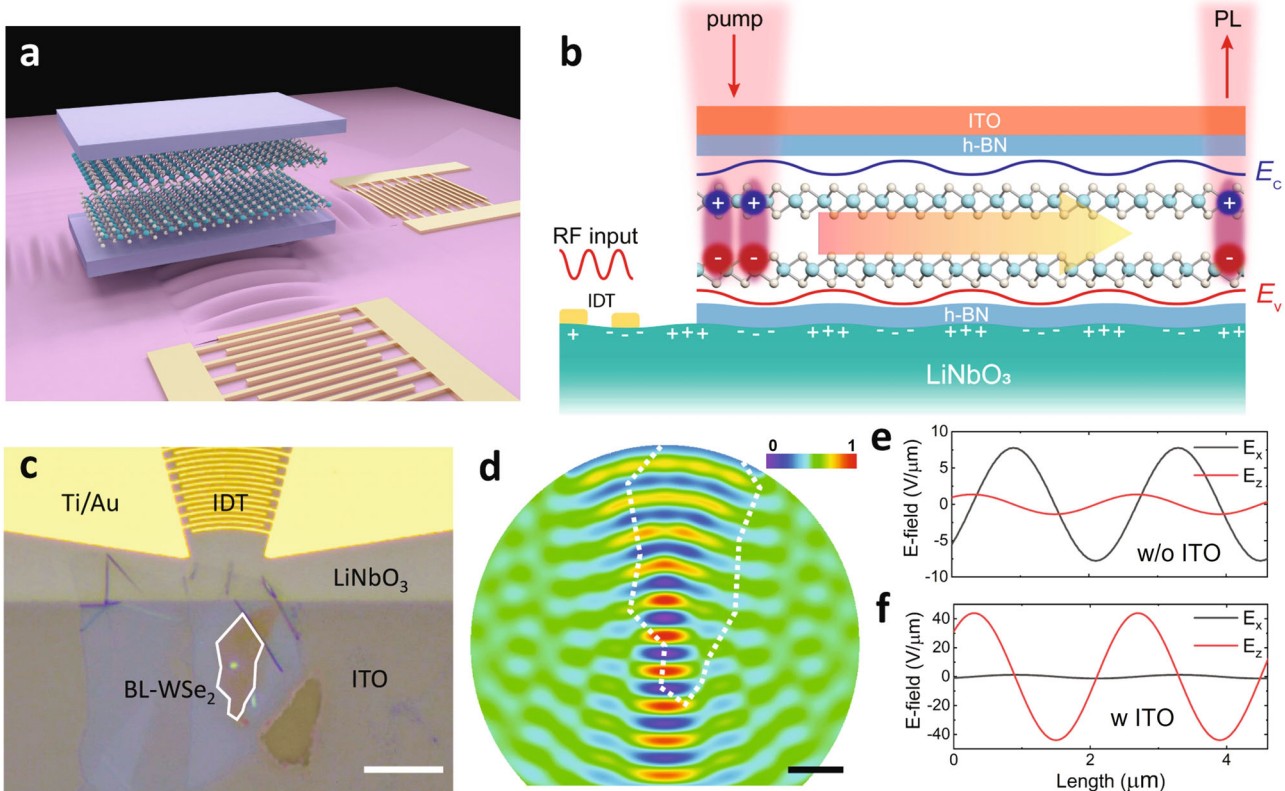

**Fig. 1 Bilayer WSe$_2$ integrated with SAW devices. a** Schematic illustration of the h-BN encapsulated bilayer WSe$_2$ stacked on SAW devices. The IDTs generate SAW to transport the excitons in different directions. **b** The propagating SAW modulates the energy of the excitons and transports them from the pump spot to the flake edge, where they recombine to generate photoluminescence. For simplicity, we only plot +z interlayer exciton in bilayer WSe$_2$. See S.I. for details. **c** Optical microscope image of the device, with the white line outlining the bilayer WSe$_2$. Scale bar, 20 μm. **d** The piezoelectric field profile of the SAW generated by the focusing IDT. The acoustic focal point has a waist of 3 μm and is at the edge of the bilayer WSe$_2$. The dotted line outlines the WSe$_2$ flake. Scale bar, 5 μm. **e, f** The piezoelectric field distribution with (**e**) and without (**f**) the top ITO electrode. The field amplitude is calculated assuming an acoustic power density of 1 mW/μm. The ITO electrode can efficiently suppress the in-plane piezoelectric field component ($E_x$), which causes undesirable exciton dissociation.

beam (Fig. 1d). Without SAW (Fig. 2a), the IXs diffuse by only 1–2 μm, consistent with previous measurement results. When SAW is turned on with power $P_s$ = 6 mW, as shown in Fig. 2b, we observe strong exciton emission at two spots on the far edge of the WSe$_2$ flake, along the SAW propagation direction and ~20 μm away from the pump spot. The two separate emission spots are at the corners (one convex and one concave) of the flake edge, where the acoustic wave is most focused (Fig. 1d). The IXs are transported at the acoustic velocity of 3500 m/s so the traveling time is less than 6 ns, shorter than their lifetime[29]. The results reveal SAW-driven transport of IXs over a device size limited distance, setting the lower bound of the propagation length to ~20 μm.

To confirm the transport is indeed from IXs, we performed spatial and energy-resolved measurements. We aligned the slit of the spectrometer with the SAW propagation direction (y-axis) and acquired spectral PL images with SAW off and on (Fig. 2c, d). The emission of the transported IXs is predominantly at the energy around 1.56 eV, which agrees with the IX energy of bilayer WSe$_2$[29,44,45]. The result is consistent with our expectation that only IXs with perpendicular dipole moment are efficiently transported by SAW. At the pump spot (Fig. 2e), the IX emission slightly decreases when SAW is turned on, presumably due to the transport of IXs by the SAW. In contrast, at the far edge of the flake (Fig. 2f), the application of SAW increased the IX emission intensity by more than two orders of magnitude. In control devices without the top ITO layer, no IX emission (see Figure 11

in S.I.) can be observed beyond the IX diffusion distance. It agrees with the expectation that the in-plane piezoelectric field, if not screened by the ITO (Fig. 1e), will dissociate the IXs to free carriers[46]. These free carriers will have a very low recombination rate in the SAW because they are spatially separated by half the acoustic wavelength (~1.4 μm in our device). Therefore, we can confirm that the emission at the edge of the flake is from SAW transported IXs.

**Power-dependent exciton transport.** We next characterize the IX transport at different SAW powers $P_s$. We find that the transported exciton density increases monotonically with the SAW power, but the trend is highly nonlinear with an activation behavior. For $P_s$ < 3.0 mW, the IX transport is negligible, and the IX emission is localized near the laser excitation spot (Fig. 3a). When $P_s$ increases to 4.5 mW, two emission spots appear at the flake edge (Fig. 3b), suggesting activation of the transport process. When $P_s$ is further increased to 6 mW, the exciton transport becomes so efficient that the emission intensity at the flake edge is already comparable to that at the pump spot (Fig. 3c). Figure 3d plots the integrated emission intensity at the flake edge, which is proportional to the transported exciton density $n_T$, as a function of $P_s$. The experimental result agrees with our theoretical model of SAW activated transport in which the transported exciton density $n_T$ is proportional to $e^{\sqrt{\frac{P_s}{P_t}}}$ at a given temperature (see S.I.). Fitting the result at 100 K gives a relatively small threshold

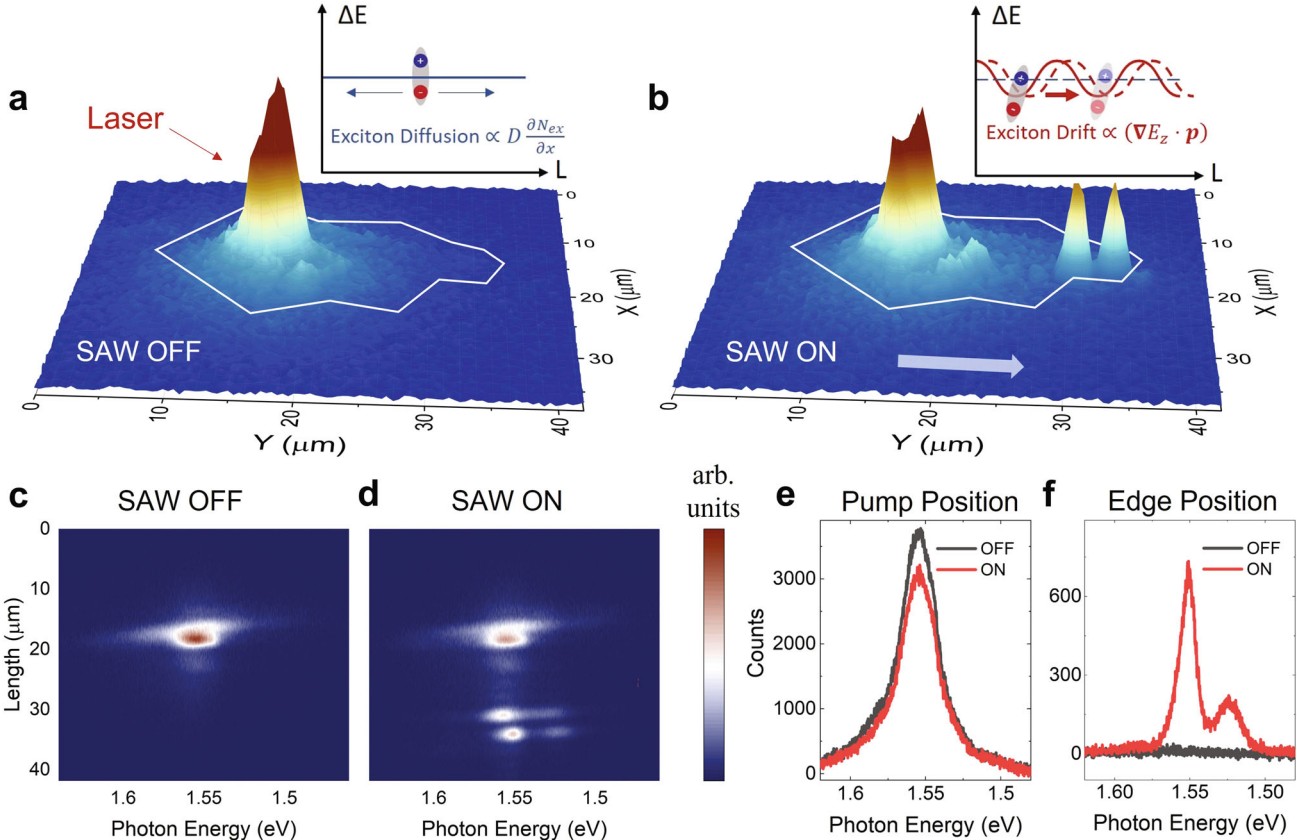

**Fig. 2 SAW-driven transport of IXs at 100 K.** Real-space PL mapping (**a**), when SAW is off, and (**b**) when SAW is on with 6 mW power. Two bright emission spots appear at the edge of the flake and the focal point of the acoustic wave (see Fig. 1d) due to the SAW-driven transport of IXs. Insets: Illustration of free diffusion and SAW-driven drift of IXs. **c**, **d** The spectral PL image at the same experimental conditions as **a**, **b**. The non-local exciton emission at the flake edge is clearly attributed to the IXs in WSe$_2$, which have an emission peak at 1.56 eV. **e** The emission spectrum at the pump position slightly decreases when SAW is turned on. **f** The emission spectrum at the flake edge position drastically increases when SAW is turned on.

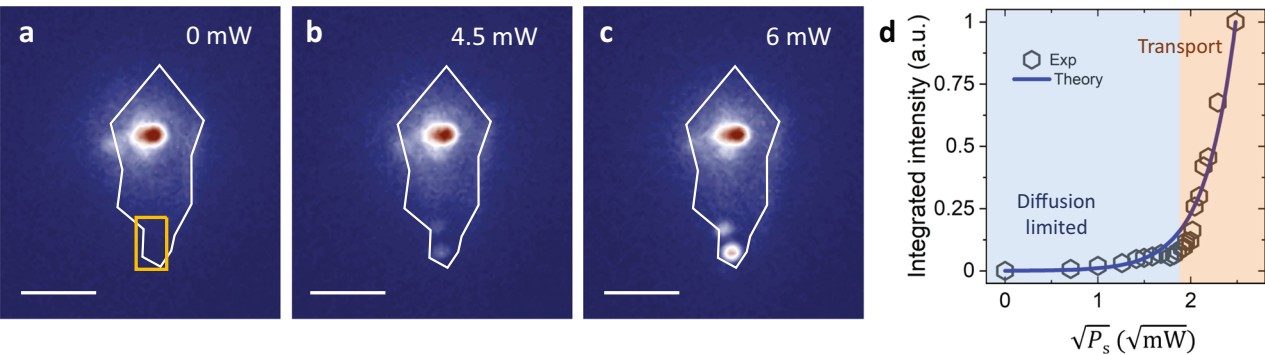

**Fig. 3 Acoustic power-dependent IX transport at 100 K.** PL images at different SAW power $P_s$ of (**a**) 0 mW, (**b**) 4.5 mW, (**c**) 6 mW. The solid white line outlines the WSe$_2$ flake. The two bright emission spots at the edge of the WSe$_2$ flake highlight the SAW-driven transport of IXs. Scale bar, 10 μm. **d** The integrated emission intensity at the flake edge (in the area indicated by the yellow box) in **a** depending on the SAW power $P_s$. The experimental data is fitted with the theoretical model that the transport exciton density exponentially depends on the square root of $P_s$. At low $P_s$, the exciton transport is diffusion-limited (blue shaded). A high $P_s$, SAW-driven transport is activated (red shaded).

power of $P_t \sim 0.1$ mW, beyond which exciton transport is activated by the SAW. Similar exponential behavior and power law have also been observed in coupled GaAs quantum wells[14], where exciton transport is impeded by disorder and defect-induced potential variations at the low SAW power limit. In TMDC such as bilayer WSe$_2$, the intrinsic defect density can be $>7.0 \times 10^{10}$ cm$^{-2}$, along with the strain-induced potential variation caused by the transfer process[40]. For efficient IX transport to happen, the SAW modulation of IX energy $\Delta U$ needs to overcome the defect and strain

induced potential variation (see S.I.), and the efficiency of exciton transport is sensitive to the material quality.

**Temperature-dependent Exciton transport.** The activation behavior motivates us to further measure SAW-driven IX transport at different temperatures. Figure 4 shows the spectral PL mapping at temperatures from 6 K to 200 K with fixed pump power $P_p = 20$ μW and SAW power $P_s = 6$ mW. In this wide

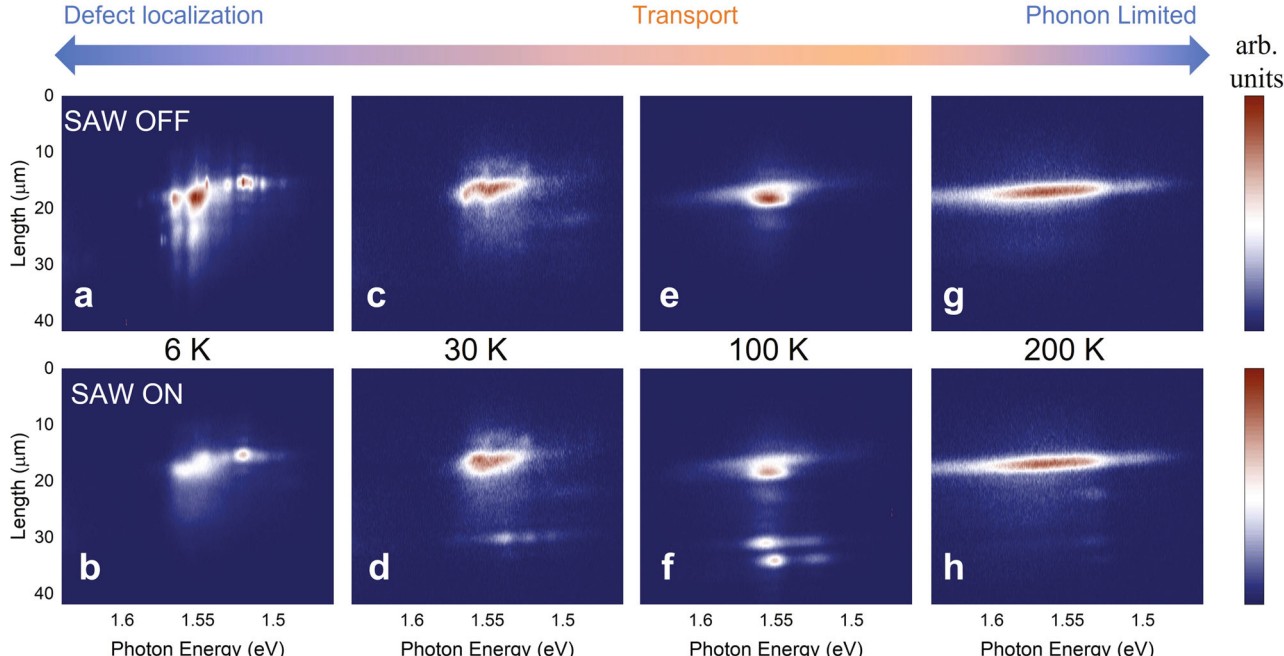

**Fig. 4 SAW-driven IX transport at varying temperatures.** The spectral PL image of exciton transport at 6-200 K with SAW off (**a**, **c**, **e**, **g**) and on (**b**, **d**, **f**, **h**). For clarity, the data at each temperature is normalized separately. **a**, **b** At low temperature (6 K), IXs show narrow peaks as they are highly localized to recombination centers such as defects or couple strongly to local vibrational modes. SAW can delocalize the IXs so to suppress the emission. However, the transported IXs remain dark because of the reduced phonon-assisted recombination at low temperature. **c**, **d** At elevated temperatures (30 K), the IXs are thermalized with broad emission. SAW transport turns on with visible emission at the flake edge. **e**, **f** At 100 K, SAW transport of IXs is most prominent as IXs have a sufficiently long lifetime and are less susceptible to defect trapping. **g**, **h** At even higher temperatures, SAW transport, although still visible, is inefficient as exciton-phonon scattering dominates and the IX population lifetime is short.

temperature range, we observe rich features showing the roles of exciton localization, SAW-driven transport, recombination, and phonon scattering. Note that the measurement at each temperature is normalized to its respective maxima for clarity. At 6 K and with SAW off, the emission spectrum shows a series of sharp resonances (Fig. 4a), which are attributed to defect emission and phonon-assisted recombination of the IXs. When SAW is on, the sharp resonances disappear and the emission intensity decreases significantly (Fig. 4b and SI). Meanwhile, IX transport at this low temperature is very weak. The suppression of IX emission can be explained by that the SAW reduces the coupling between excitons and recombination centers such as defects. When the temperature increases to 30 K, the sharp exciton resonances disappear (Fig. 4c). At this elevated temperature range, the excitons are thermalized into exciton gas, leading to weaker couplings with the local defects. As a result, the SAW-driven transport starts to make the PL at the flake's edge observable (Fig. 4d). When the temperature reaches 100 K, the IX emission becomes more than two times brighter at the pump spot (Fig. 4e) than at 30 K. This is because indirect IX recombination becomes more efficient when the thermal phonon density is high. The SAW-driven transport is also the most prominent at this temperature (Fig. 4f) because the defects and disorders have a less trapping effect on the thermalized excitons and the exciton population lifetime is still sufficiently long. At 200 K (above the Debye temperature), the thermal phonon population is high. As a result, IX emission has a broad spectrum and high intensity without SAW (Fig. 4g). However, the IX transport is impeded due to strong exciton-phonon scattering and decreased exciton population lifetime (Fig. 4h). Overall, our systematic measurements have revealed three regimes of IX behaviors and the transition between them. At the lowest temperatures, the IXs are highly localized and their

transport is diffusion-limited. At intermediate temperatures, the IXs are bright with phonon-assisted recombination. At the same time, they still have a long lifetime, allowing for efficient SAW-driven transport over a long distance. At further elevated temperatures, strong phonon scattering decreases exciton mobility and lifetime, preventing efficient transport. Nevertheless, thanks to the strong SAW modulation, even at room temperature, we still observe a SAW-driven transport distance of ~2.0 μm (see Supplementary Fig. 8).

## Discussion
In conclusion, we have demonstrated that SAW is an efficient, contact-free approach to transport IXs in bilayer WSe$_2$ over a distance far beyond the diffusion length. The SAW-driven transport happens when the SAW modulation of IX energy can overcome the local potential variation and defect traps. Since the transport distance depends on the exciton lifetime of the optically active material and the acoustic velocity of the substrate, using TMDCs with longer lifetimes and piezoelectric substrate with higher acoustic velocity can lead to a much longer transport distance. The contact-free transport driven by the acoustic wave that is launched remotely also preserves the high quality of the materials and prevents undesired effects induced by local gates[35-37,39]. Although the maximal transport distance is reached at the temperature of 100 K in the current device, further improvement of the material quality and interface cleanness will make efficient room-temperature operation possible. We note that SAW is a universal approach to control excitons with both of its piezoelectric field and strain field can be utilized to manipulate and transport excitons in many other 2D material systems. SAW can also be guided and circulated in phononic circuits and resonators[47,48], which will afford rich functionality and flexibility.

## Methods

**SAW-exciton device fabrication**. The WSe$_2$ device was fabricated by the standard polymer-assisted (PC) pick-up method. The 2D flakes of hBN and WSe$_2$ (HQ graphene) were exfoliated to a 90 nm SiO$_2$/ Si substrate and then picked up using PC/PDMS (Sylgard 184) stamps. The bilayer WSe$_2$ flakes were identified by their optical contrast and then confirmed with photoluminescent measurements. The IDTs were patterned on a y-cut LiNbO$_3$ wafer (MTI) using ebeam lithography. The IDT is aligned to generate SAW propagating along the z-axis of the LiNbO$_3$ wafer. Layers of 12 nm chromium and 120 nm gold were deposited using an ebeam evaporator under high vacuum. Aligned transfer was then done by a home-built transfer stage with high accuracy so that the WSe$_2$/hBN heterostructure was precisely aligned at the focus region of the curved IDTs. PC residue was then removed using chloroform for 1 h and followed by a 5 min rinsing in the IPA. Finally, aligned ebeam writing was performed and 50 nm indium-tin-oxide (ITO) was deposited by a sputtering system (Evatec LLS EVO) under O$_2$ condition. After deposition, the device was annealed in atmosphere at 300 °C for 5 min to improve the conductivity of the ITO film, which can ensure a better screening of the in-plane field.

**Measurement setup**. A continuous-wave He-Ne laser (633 nm) was used to excite the excitons in bilayer WSe$_2$. With an objective lens (NA = 0.42), the laser beam was focused with a diffraction-limited spot size of about 1 μm. The sample was mounted in a cryostat (Montana Instrument) with an optical window for optical access. Meanwhile, the RF signal was generated by a vector network analyzer (VNA) (Agilent E8362B) and then coupled to the wire-bonded device inside the cryostat. The calibration kits (Keysight 8052D) were used to de-embed the system so that the IDT resonance can be resolved. The interlayer exciton emission was acquired with a spectrometer (Princeton Instrument) after the laser line was removed with a 633-notch filter (Thorlabs). To image the spatial transport of the exciton, the center wavelength of the spectrometer was set to be 0 so it functioned like a mirror. The filtered signal was then collected by a cooled camera (Pixis 400) operating at −70 °C to improve the signal-to-noise ratio. To collect the spectrum results, we set the center wavelength to 790 nm, which covered the emission spectrum of the interlayer exciton. To perform the spectral-PL imaging measurement of the exciton transport, the transport direction was aligned with the slit of the spectrometer, while the x-axis of the CCD camera displayed the spectrum information. The y-axis signal gave the spatial emission of the bilayer devices.

**Reporting summary**. Further information on research design is available in the Nature Research Reporting Summary linked to this article.

## Data availability

The data that support the findings of this study are available from the corresponding author upon reasonable request.

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

## Acknowledgements

This work was supported by the National Science Foundation (NSF Award No. EFMA 1741656 and NSF DMR-1719797). Part of this work was conducted at the Washington Nanofabrication Facility/Molecular Analysis Facility, a National Nanotechnology Coordinated Infrastructure (NNCI) site at the University of Washington with partial support from the National Science Foundation via awards NNCI-1542101 and NNCI-2025489.

## Author contributions

R.P., M.L., X.X., T.C. conceived the research. R.P. and A.R. fabricated the devices, performed the measurements. Y.Y. and T.C. performed theoretical analysis. R.P. analyzed the data. J.Z., C.W., S.L. assisted device fabrication and data analysis. T.T. and K.W. provided the hBN material. R.P., M.L., X.X., and C.T. co-wrote the manuscript with contributions from all authors.

## Competing interests

The authors declare no competing interests.
