## [Peer Review File · Nature Communications]

Reviewers' Comments:

Reviewer #1:

Remarks to the Author:

Authors have replied to my previous comments and especially showed the simulation result for microsecond long lifetime, the possibility of Vally hall effect and high-temperature result. I am glad to recommend its publication now.

Reviewer #2:

Remarks to the Author:

The authors revised the manuscript based on the recommendations of the reviewers. All important technical points and comments related to the presentation have been properly addressed.

This reviewer believes the reported results are indeed of relevance in the field of acoustic technologies and 2D materials and, therefore, recommends publication in *Nature Communications*.

The Authors' Response to Reviewers' Comments

Reviewer #1:

Authors have replied to my previous comments and especially showed the simulation result for microsecond long lifetime, the possibility of Vally hall effect and high-temperature result. I am glad to recommend its publication now.

Our response: We thank the reviewer's positive comments and a strong recommendation for publication.

Reviewer #2:

The authors revised the manuscript based on the recommendations of the reviewers. All important technical points and comments related to the presentation have been properly addressed.

This reviewer believes the reported results are indeed of relevance in the field of acoustic technologies and 2D materials and, therefore, recommends publication in Nature Communications.

Our response: We thank the reviewer for the positive evaluation of our results and the recommendation for the publication.